# Dynamic Modeling of Poultry Litter Composting in High Mountain Climates Using System Identification Techniques

Alvaro A. Patiño-Forero *,†, Fabian Salazar-Caceres †, Harrynson Ramirez-Murillo *,†, Fabiana F. Franceschi †, Ricardo Rincón † and Geraldynne Sierra-Rueda †

Faculty of Engineering, Universidad de La Salle, Cra. 2 No. 10-70, Bogotá 111711, Colombia; jfsalazar@unisalle.edu.co (F.S.-C.); ffranceschi@unisalle.edu.co (F.F.F.); rrinconb@unisalle.edu.co (R.R.); gsierra10@unisalle.edu.co (G.S.-R.)

* Correspondence: alapatino@unisalle.edu.co (A.A.P.-F.); haramirez@unisalle.edu.co (H.R.-M.)
† These authors contributed equally to this work.

## Abstract

Poultry waste composting is a necessary technique for agricultural farm sustainability. Composting is a dynamic process influenced by multiple variables. Humidity and temperature play fundamental roles in analyzing its different phases according to the environment and composting technique. Current developments for monitoring these variables include automation via intelligent Internet of Things (IoT)-based sensor networks for variable tracking. These advancements serve as efficient tools for modeling that facilitate the simulation and prediction of composting process variables to improve system efficiency. Therefore, this paper presents the dynamic modeling of composting via forced aeration processes in high-mountain climates, with the intent of estimating biomass temperature dynamics in different phases using system identification techniques. To this end, four dynamic model estimation structures are employed: transfer function (TF), state space (SS), process (P), and Hammerstein–Wiener (HW). The and model quality, fitting results, and standard error metrics of the different models found in each phase are assessed through residual analysis from each structure by validation with real system data. Our results show that the second-order underdamped multiple-input–single-output (MISO) process model with added noise demonstrates the best fit and validation performance.

**Keywords:** aerobic composting; composting monitoring; dynamic modeling; system identification; wireless sensor network

## 1. Introduction

Global poultry production generates substantial nutrient-rich waste if mismanaged, with environmental risks involving soil, groundwater, and greenhouse gas emissions [1]. Composting transforms poultry litter into organic fertilizer, mitigating these impacts while enhancing soil fertility for sustainable agriculture [2]. In high-mountain climates such as Cundinamarca, Colombia (2835 m above sea level), low temperatures (7–19 °C) and fluctuating moisture levels hinder the achievement of optimal composting conditions such as the thermophilic temperatures (≥50 °C) necessary for effective pathogen elimination [3]. This study employs system identification techniques with IoT-based monitoring to model and optimize poultry litter composting dynamics under these challenging conditions.

The surge in global meat demand has intensified poultry production, resulting in substantial waste from egg production that requires sustainable management [4]. Aerobic composting using forced-aeration static piles effectively transforms poultry litter into

fertilizer, mitigating emissions, soil erosion, and dependence on synthetic fertilizers [5,6]. The key parameters of temperature, moisture, and aeration govern microbial activity and process efficiency; however, their complex interactions challenge traditional modeling [7–9]. In high-mountain environments, these challenges are amplified by environmental variability, necessitating advanced adaptable modeling approaches [1].

High-mountain composting faces unique constraints, as low ambient temperatures and variable moisture impede the thermophilic phases that are critical for organic matter stabilization [3,7]. Insufficient aeration risks anaerobic conditions, whereas excessive aeration may reduce efficiency by preventing thermophilic temperatures [9]. In addition, variations in moisture levels tied to the makeup of the substrate make microbial activity more difficult to manage [8]. Traditional models focusing on heat and mass transfer are computationally intensive and typically tailored to specific materials, which limits their usefulness in varied settings [10,11].

In Colombia, an experimental study was conducted at high altitude to evaluate composting strategies under mountain climate conditions [12]. The research showed that low temperatures and small-scale operations hinder the development of thermophiles and effective sanitization. The application of bioactivators and bokashi has improved compost quality and reduced pathogen levels. While this study offers a comprehensive experimental evaluation, it does not incorporate modeling approaches or system identification techniques.

Recent advancements in composting modeling include statistical methods such as response surface optimization [13] and artificial neural networks (ANN) [14–16], which have been extended by hybrid mechanistic data-driven models and advanced machine learning approaches to enhance process optimization. For instance, a hybrid modeling approach integrating mechanistic and data-driven methods for fermentation process optimization was reviewed in [17], with an emphasis on system identification for bioreactor control. Similarly, a deep hybrid model for bioreactor systems combining first principles with neural networks was developed in [18], resulting in enhanced control accuracy Advanced ML approaches such as random forest [19], deep learning (e.g., transformers), and reinforcement learning offer improved predictive accuracy for complex biological processes [11,20]. Concurrently, system identification methods have become relevant in modeling biological and environmental processes. Hybrid differential equations have shown good predictive capabilities when employed for simulating water systems, as demonstrated in [21]. Similarly, a system identification approach for real-time control of anaerobic digestion was described in [22], emphasizing its flexibility. Industry 4.0 technologies such as IoT and AI can enable real-time monitoring and control to provide enhanced agricultural efficiency [23–26]. These advancements highlight the potential of such data-driven approaches, including the one proposed in this study to address the complexities of composting in high-mountain climates.

Compared to traditional models, ML techniques exhibit higher predictive accuracy when simulating compost dynamics. In addition to this enhanced accuracy, ML approaches offer advantages such as faster computation, lower processing costs, and reduced demand for experimental resources and human labor, making them highly suitable in organic waste composting systems [27]. For example, [28] reported that the use of AI and ML tools has enabled the optimization of composting parameters, prediction of compost maturity, monitoring of moisture content in industrial-scale systems, estimation of compost enzymatic activity, and classification of compost maturity using theoretical, analytical and statistical methods [29].

In [30], the authors forecast humic acid content using a backpropagation ANN, while [31] found that ANN models provided the best performance for modeling the composting process. In [32], the authors compared the performance of an ANN and a response surface methodology (RSM) model in optimizing compost maturity parameters. Their study demonstrated that while both models were efficient, ANN presented an advantage compared to RSM. In another study [33], the authors employed six ML methods to develop models for predicting the germination index (GI) during manure composting, finding that the random forest (RF) and extra trees (ET) models presented the best predictive performances for GI. Similarly, [34] tested seven ML models to predict the humification index (HI) during composting, finding that the gradient boosting regression tree technique provided the best performance for modeling HI. The authors also identified that the C/N ratio and aeration rate were the most influential variables in HI modeling. Finally, [35] presented a sensor-based ML system to predict compost maturity and monitor gas emissions in real time. Using environmental data and ten composting datasets, models such as XGBoost and CatBoost achieved high predictive accuracy. The tested approach was found to enhance waste management efficiency, transparency, and sustainability. While the above studies demonstrate the relevance of ML in composting modeling, there are still challenges related to data requirements and model complexity, highlighting the need for alternative approaches [10,11].

System identification offers a practical and data-driven approach for the modeling of composting in high-mountain climates, yet its application in such contexts remains underexplored [19,36]. The integration of system identification with IoT data provides a resource-efficient alternative that can model composting dynamics without the need for extensive biochemical characterization [37,38].

Despite these advances, a gap remains in modeling composting processes for high-mountain conditions using adaptable methods. Several methods require extensive knowledge of biochemical processes, which consider large datasets or do not respond well to environmental variability. In contrast, this study uses IoT data to develop and test dynamic models for controlling temperature in a forced-aeration poultry litter composting system in Cundinamarca, Colombia. The resulting models help to improve process efficiency and provide a scalable, environmentally friendly solution for waste management in challenging environmental conditions.

## 2. Materials and Methods

This section describes the experimental setup and methods used to develop dynamic models for temperature control in poultry litter composting systems under high-mountain climate conditions. The study was conducted at an automated composting pilot plant in La Vega, Cundinamarca, Colombia which is equipped with IoT-based monitoring and control systems. The facility enables forced aeration and temperature regulation through a heat exchanger, supported by real-time data acquisition from calibrated environmental sensors. Two composting experiments were carried out using poultry manure and sawdust mixtures, with sensor data collected at regular intervals to capture process dynamics. The collected data were used to develop and compare several system identification models aiming to represent the biopile's thermal behavior efficiently under variable environmental and operational conditions.

### 2.1. Automated Composting Plant

This study was conducted at a research and training center located in La Vega ($4°52'18.932''$ N, $74°25'6.54''$ W), Cundinamarca, Colombia, at the Alto del Vino (2835 m above sea level) on the eastern mountain range to the west of Bogotá. The poultry pro-

duction farm experiences average daytime temperatures of approximately 19 °C, which decrease to around 7 °C during the night [1]. The composting plant, housed in a greenhouse with brick and cement flooring, comprises four cubicles (i, j, k, l), each with a 4.3 m³ capacity, as shown in Figure 1. Each cubicle uses 4.0-inch perforated PVC pipes for aeration, supplied by an industrial fan (c) controlled by manual valves (n, o, p, q). The fan assembly includes a damper valve (d) and a gas burner (b) with a heat exchanger equipped with a solenoid valve and an ignition pilot. A variable speed drive adjusts airflow. Three sensors monitor the process: a temperature sensor (e) at the heat exchanger outlet, a combined temperature and oxygen sensor (g) inside the biopile, and a humidity sensor (h), also in the biopile. All sensors are connected to an IoT-based control system (f) for both manual and remote operation. The thermophilic phase was maintained at approximately 40 °C for 7 days, constrained by the high-mountain climate's low ambient temperatures (7–19 °C) and the heat exchanger's capacity, preventing the system from reaching the ≥55 °C threshold recommended for pathogen elimination [3]. This phase is aligned with cold-climate composting studies, where lower temperatures suffice to stabilize organic matter.

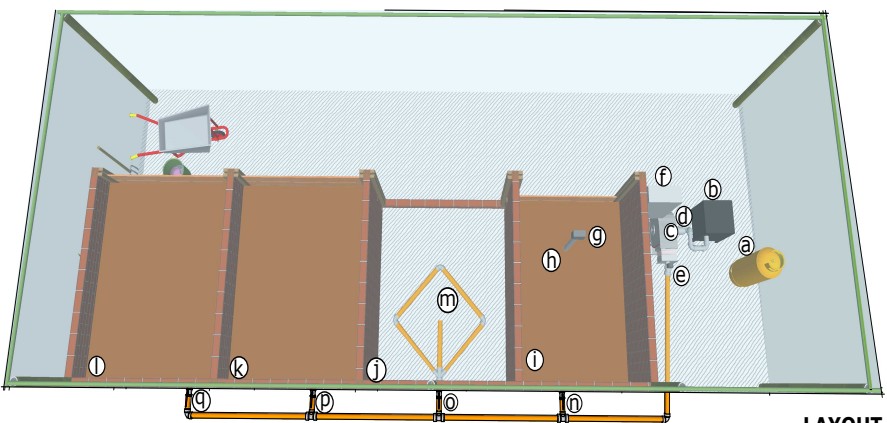

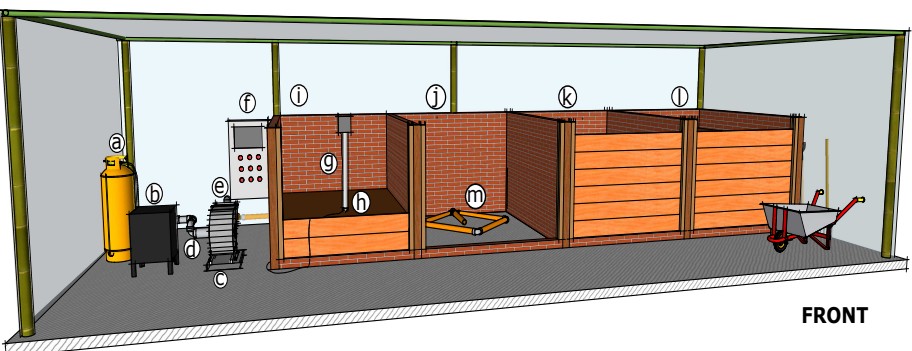

**Figure 1.** Process and instrumentation diagram: (a) gas pipe, (b) heat exchanger, (c) blower, (d) damper, (e) temperature sensor, (f) system control, (g) temperature and oxygen sensors, (h) moisture sensor, (i–l) modules, (m) floor aeration pipe, hand valves (n–q).

## 2.2. Control and Communications Architecture

Figure 2 shows the control and communications architecture, which is composed of an industrial Arduino that acquires all analog signals of the system (biopile temperature, air temperature, moisture, and damper position). The acquired signals are sent by serial communication to an industrial Raspberry Pi that includes an integrated human–machine interface where the plant commissioning operations are performed (ignition of the gas heat

exchanger, output control, control of the frequency inverter cfw 300b, and control of the damper valve). The Raspberry Pi is connected to a modem to allow internet access via GSM/GPRS.

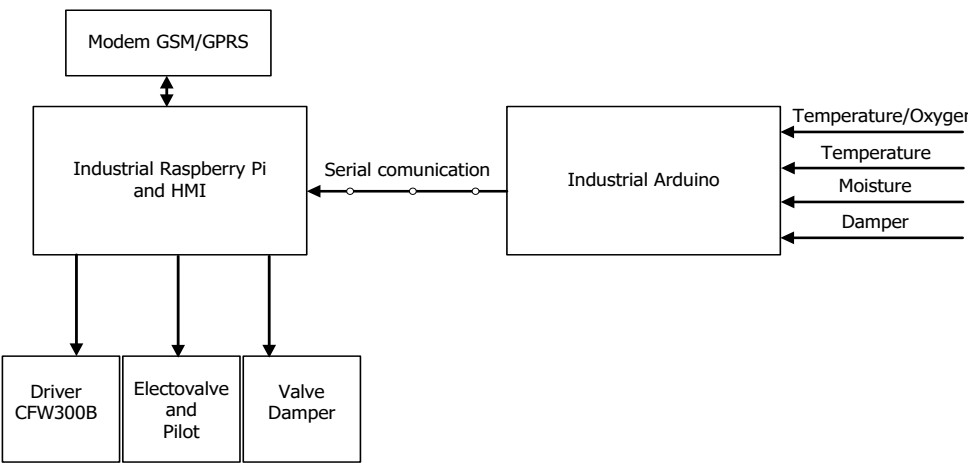

**Figure 2.** Technology architecture for control and communications.

The experimental procedure at the pilot plant began with manual collection of poultry manure from the barn, which was subsequently placed in each cubicle. Manure–pine saw-dust mixtures were prepared to control excess moisture and enhance biomass porosity [3]. Figure 3a shows the four cubicles loaded and ready for operation. Before starting, two pipeline valves were opened to aerate two cubicles simultaneously and a temperature and oxygen sensor (LSI Lastem) and moisture sensor were inserted into the biomass, as shown in Figure 3c. The operator verified the process variables (Figure 3d) and activated the heat exchanger and industrial fan via the inverter. Aeration was conducted separately for cubicles i and j (Experiment 2) and cubicles k and l (Experiment 1) over 15 days each, with a 5-min sampling interval used to capture rapid temperature changes during the thermophilic phase while balancing computational efficiency [7].

The plant operated 12 h nightly due to low temperatures (7–19 °C). Table 1 lists the initial conditions. This decision was based on local environmental conditions, where daytime temperatures in the region rise to around 19 °C and with peaks of up to 23 °C. The plant is located inside a greenhouse that retains heat and minimizes sharp daytime temperature drops; these conditions reduced the need for additional heating during daylight hours, allowing for efficient thermal management with nighttime operation only.

Experiment 1 collected 4302 data points related to biomass temperature, substrate moisture, and hot air temperature, while Experiment 2 collected 3077 data points at the same sampling rate, which were used for model validation. All sensor data were preprocessed using a moving average filter (window size = 5 samples) to reduce high-frequency noise, and outliers exceeding ±3 standard deviations were removed. Data acquisition was implemented using Node-RED with parallel storage in MariaDB and Firebase databases.

Substrate moisture was measured using an FDS-100 resistive soil moisture sensor (probe diameter: 3 mm, output signal: 4–20 mA, operating voltage: 7–24 V, current consumption: 3–5 mA, cable length: 1.5 m). Biomass temperature and oxygen concentration were monitored with an EXP421 LSI Lastem sensor (thermistor-based for temperature and electrochemical for oxygen; temperature range: from −40 to −70 °C, accuracy: ±0.5 °C; oxygen range: 0–25 %, accuracy: ±0.3 %). The hot air temperature inside the combustion chamber was measured using a Type-K thermocouple connected to a gas burner unit. All

sensors were calibrated weekly to minimize measurement drift. Measurements were taken every 5 min (sampling frequency: 0.0033 Hz) throughout the experimental periods.

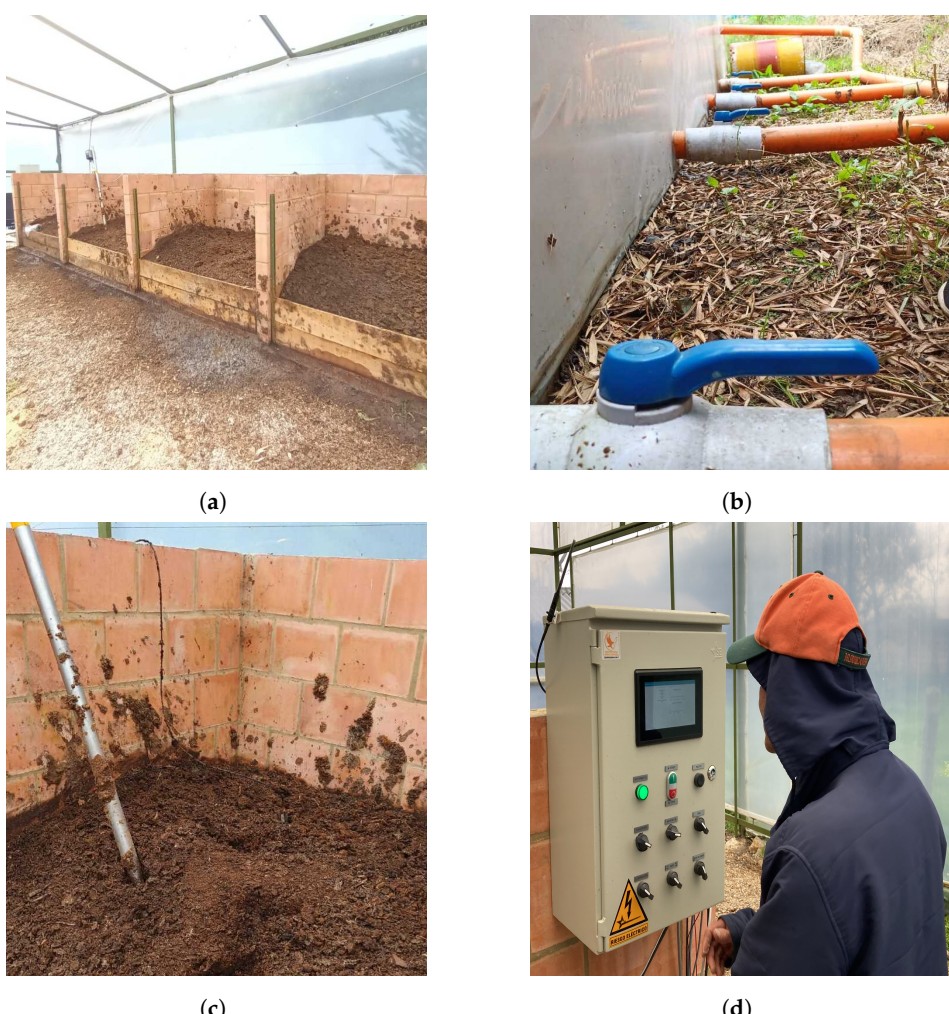

**Figure 3.** Commissioning of the automated composting plant with forced aeration: (**a**) cubicles loaded with chicken manure, (**b**) forced aeration techniques, (**c**) sensor implementation in the biopile, (**d**) plant verification and commissioning.

**Table 1.** Characteristics of the experiments.

|  | **Experiment 1** | **Experiment 2** |
|---|---|---|
| Biopile Composition Manure (%) | 63 | 63 |
| Sawdust (%) | 37 | 37 |
| Humidity (%) | 100 | 100 |
| Aeration (L/m) | 1.092 | 1.092 |
| Experiment time (h) | 12 | 12 |

Selection of the Estimation Model

After collecting the data of the measured variables [39], the input (humidity, heater temperature, and air flow) and output (biomass temperature) data were preprocessed and the system was classified as MISO [40]. Within the theory of system identification, it is recommended to start with a mathematical relationship of inputs and outputs and then move on to more complex structures [41]. Therefore, three structures were initially selected: transfer function (TF), state space (SS), and process (P); then, a nonlinear Hammerstein–

Wiener (HW) structure was selected. Another important characteristic to consider in system identification is the model order; accordingly, the study began with a review of classical energy and mass balance equations found in the literature describing the composting process. It was observed that most of these models are first-order; for this reason, the modeling work was started with first- and second-order models. Model estimation was performed using 50% of the measured data from Experiment 1. The dataset used for this purpose started at data point 1235, as the humidity from that point onward remained within the range of 50% to 60%, ensuring that the composting process had properly commenced.

### 2.3. Model Estimation by Transfer Function

Model estimation using a discrete transfer function was applied to the composting process, as it enables straightforward implementation of a simulation model or future predictions of plant behavior [42]. For this purpose, the model was implemented in Matlab using the `tfest` function, which applies the output-error (OE) polynomial algorithm [43]. The values of the three inputs and one output were parameterized using a sampling time of 300 s. Among the evaluated models, the second-order model provided the best response in terms of system representation. The identified transfer functions are presented in Equations (1)–(3):

$$\frac{BT(z)}{HT(z)} = \frac{0.001204z^{-1}}{1 - 0.569z^{-1} - 0.431z^{-2}}, \tag{1}$$

$$\frac{BT(z)}{H(z)} = \frac{0.001929z^{-1}}{1 - 0.9858z^{-1} - 0.01075z^{-2}}, \tag{2}$$

$$\frac{BT(z)}{F(z)} = \frac{0.0003272z^{-1}}{1 - 0.0197z^{-1} - 0.9791z^{-2}}, \tag{3}$$

Respectively corresponding to the inputs of $HT(t)$ (heater temperature), $H(t)$ (humidity), and $F(t)$ (flow) used in the estimation. The system's sole output is $BT(t)$ (biopile temperature).

### 2.4. State-Space Model Estimation

The state-space model is a mathematical representation of a system in which the inputs and outputs are related through a set of first-order differential equations. To estimate the discrete-time model of the composting process, a sampling time $T_s$ was defined and the model order was determined using the `ssest` function in Matlab, which employs the canonical variate algorithm. The discrete-time state-space representation of the system is provided by Equations (4) and (5) [44]:

$$\mathbf{x}(t+1) = A\mathbf{x}(t) + B\mathbf{u}(t) + Ke(t), \tag{4}$$

$$y(t) = C\mathbf{x}(t) + D\mathbf{u}(t) + e(t), \tag{5}$$

where $A, B, C, D$, and $K$ are matrices in the state space of the appropriate dimensions, $\mathbf{u}(t)$ is the input vector, $y(t)$ is the output, $e(t)$ is the perturbation, and $\mathbf{x}(t)$ is the state vector. The estimated matrices of this type of representation are presented below in Equations (6)–(10):

$$A = \begin{bmatrix} 0 & 1 \\ -0.9859 & 1.986 \end{bmatrix}, \tag{6}$$

$$B = \begin{bmatrix} 0.0005328 & 0.002377 & 0.000192 \\ 0.0005416 & 0.00235 & 0.0001874 \end{bmatrix}, \tag{7}$$

$$C = \begin{bmatrix} 1 & 0 \end{bmatrix}, \tag{8}$$

$$D = \begin{bmatrix} 0 & 0 & 0 \end{bmatrix}, \tag{9}$$

$$K = \begin{bmatrix} 0.1613 \\ 0.1607 \end{bmatrix}. \tag{10}$$

An observable canonical form was realized, representing the state-space system in a reduced-parameter structure where specific elements of the matrices are fixed to zero or one. The canonical indices are identifiable in matrices $A$ and $B$. This model enables the estimation of two perturbation coefficients, denoted by $K$; the $D$ matrix is a zero vector, indicating that there is no direct feed-through from the input to the output.

### 2.5. Model Estimation by Process Structure

The process model represents poultry litter composting dynamics using a continuous-time transfer function, capturing heat transfer and microbial degradation in the biopile [45]. Key parameters include $K_p$ (static gain, reflecting the steady-state temperature response to inputs such as aeration), $T_w$ (inverse of the natural frequency, indicating the speed of temperature changes), and $\zeta$ (the damping coefficient, representing microbial activity stabilization) [46]. These parameters simplify modeling for high-mountain climates, where temperature control is critical. The model's simplicity allows for estimation of delay and for coefficients that are interpretable as poles and zeros, supporting first- to third-order models with real or complex poles [47]. Noise (white or colored) is included in the output, as shown in the discrete-domain input–output relationship in Equation (11):

$$Y(z) = G(z)U(z) + H(z)E(z) \tag{11}$$

where $Y(z)$ is the output corresponding to $BT(t)$, $G(z)$ is the discrete transfer function, $U(z)$ represents the input or inputs, $E(z)$ is the error (white Gaussian noise), and $H(z)$ is the noise sensitivity function. Model estimation was performed taking into account two dominant complex poles and a second-order underdamped system, leading to the structure presented in Equation (12), which is expressed in the frequency domain with $z = e^{j\omega T_s}$:

$$G(z) = \frac{K_p z^{-2} + 2K_p z^{-1} + K_p}{z^{-2}\dfrac{T^2 - 4T\omega^2 - 4T^2\omega\zeta}{T^2} + z^{-1}\dfrac{2T^2 - 6T\omega^2}{T^2} + \dfrac{16T\omega^4 + 4T\omega\zeta T^3 + T^4}{T^4}}. \tag{12}$$

The resulting model structure for a multiple-input–single-output (MISO) system with three inputs and one output is presented in Equation (13):

$$Y = G_{11}(z)U_1(z) + G_{12}(z)U_2(z) + G_{13}(z)U_3(z) \tag{13}$$

where $G_{11}(z)$, $G_{12}(z)$, and $G_{13}(z)$ are shown in Equations (14), (15), and (16), respectively:

$$G_{11}(z) = \frac{1.4z^{-2} + 2.8z^{-1} + 1.4}{-114299.8z^{-2} + 1.5z^{-1} + 6851543}, \tag{14}$$

$$G_{12}(z) = \frac{0.15327z^{-2} + 0.3065z^{-1} + 0.15327}{-3288.1z^{-2} + 1.42z^{-1} + 10793654.6}, \tag{15}$$

$$G_{13}(z) = \frac{-0.03574z^{-2} - 0.07148z^{-1} - 0.03574}{3.1877e^{10}z^{-2} - 1783.4z^{-1} + 1.0161e^{21}}. \tag{16}$$

For estimation of the noise sensitivity function, an autoregressive moving average (ARMA) structure is used for the perturbation model, as shown in Equation (17):

$$y(z) = G(z)U(z) + \frac{N_{no}(z)}{D_{no}(z)}E(z) \tag{17}$$

where $N_{no}$ and $D_{no}$ are provided by Equations (18) and (19), respectively:

$$N_{no}(z) = \frac{-2.21z^{-2} + 8z^{-1} + 10.21}{90000z^{-2} + 180000z^{-1} + 90000}, \tag{18}$$

$$D_{no}(z) = \frac{-2.636z^{-2} + 8z^{-1} + 10.63}{90000z^{-2} + 180000z^{-1} + 90000}. \tag{19}$$

ARMA models are autoregressive models that do not incorporate exogenous inputs, making them well suited for modeling processes based solely on time series data. They include a polynomial component that accounts for the moving average of the noise [48]. In this particular case, a second-order ARMA model was estimated, resulting in a total of twelve model coefficients.

### 2.6. Hammerstein–Wiener (HW) Model Estimation

The HW model implemented for system identification combines nonlinear input/output blocks with a linear transfer function, which can model interactions between aeration, moisture, and temperature in high-mountain biopiles [49]. As shown in Figure 4, the structure includes a nonlinear input block $f(u(t))$, a linear transfer function $B(z)/F(z)$, and a nonlinear output block $g(x(t))$.

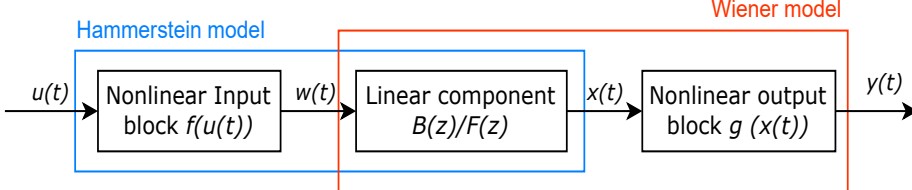

**Figure 4.** Hammerstein–Wiener (HW) structure.

This enables accurate representation of the nonlinear dynamics involved in poultry litter composting, where:

- $f(\cdot)$ is a smooth nonlinear function that transforms the input signal $u(t)$ into $w(t) = f(u(t))$ as the output of the nonlinear block.
- The linear state space composed of matrices $A, B, C, D$ of appropriate sizes can be taken to a transfer function realization provided by $\frac{B(z)}{F(z)}$ that transforms $w(t)$ into $x(t)$.
- $x(t)$ is an internal variable that represents the output of the linear block and has the same dimension as $y(t)$.
- $g(\cdot)$ is a smooth nonlinear function that maps the output of the linear block $x(t)$ to the output of the system $y(t)$ as $y(t) = g(x(t))$.

For $ny$ outputs and $nu$ inputs, the linear block is a transfer function matrix containing inputs, as shown in Equation (20):

$$\frac{B_{j,i}(z)}{F_{j,i}(z)} \tag{20}$$

where $j = 1, 2, \ldots, ny$ and $i = 1, 2, \ldots, nu$. In order to use this structure, it is necessary to specify $nb$ (number of zeros), $nf$ (number of poles), and $nk$ (delay from input to output in terms of samples) [50]. Finally, the method was implemented in the software by excluding

input and output nonlinearities, resulting in a linear transfer function. All corresponding parameters are presented in Equations (21)–(29):

$$nb = \begin{bmatrix} 2 & 2 & 2 \end{bmatrix}, \tag{21}$$

$$nf = \begin{bmatrix} 2 & 2 & 2 \end{bmatrix}, \tag{22}$$

$$nk = \begin{bmatrix} 2 & 2 & 2 \end{bmatrix}, \tag{23}$$

$$B = \begin{bmatrix} 0 & 0.0576 & -0.0566 \\ 0 & 0.3285 & -0.3253 \\ 0 & -0.0025 & 0.0025 \end{bmatrix}, \tag{24}$$

$$B_{free} = \begin{bmatrix} 1 & 1 & 1 \end{bmatrix}, \tag{25}$$

$$F = \begin{bmatrix} 1 & -1.3046 & 0.3057 \\ 1 & -0.4585 & -0.5370 \\ 1 & -1.7170 & 0.7176 \end{bmatrix}, \tag{26}$$

$$F_{free} = \begin{bmatrix} 1 & 1 & 1 \end{bmatrix}, \tag{27}$$

$$InputNonlinearity = \begin{bmatrix} 3 \times 1 & \texttt{idUnitGain} \end{bmatrix}, \tag{28}$$

$$OutputNonlinearity = \begin{bmatrix} 1 \times 1 & \texttt{idUnitGain} \end{bmatrix}, \tag{29}$$

where the terms *InputNonlinearity* and *OutputNonlinearity*, specified as `idUnitGain` in the software implementation, correspond to identity mappings in the mathematical model; that is, the nonlinear functions $f(\cdot)$ and $g(\cdot)$ are defined as in Equation (30):

$$f(\mathbf{u}(t)) = \mathbf{u}(t), \quad g(\mathbf{x}(t)) = \mathbf{x}(t), \tag{30}$$

which implies that the input and output signals are passed through the system without any nonlinear transformation. In this case, both blocks act as identity functions, meaning that the overall system behaves as a purely linear transfer function model. This simplification is often used to validate the linear dynamics of the system before introducing nonlinear components in more complex identification procedures.

The biggest strength of this approach is how flexible it is and how well it fine-tunes its settings. It starts with a basic model such as a linear state-space or transfer function, then uses a nonlinear unknown set of building blocks to tweak the model's parameters. This method helps to capture real-world input–output behaviors such as actuator saturation and hysteresis. The data are fitted much better by modeling each unique behavior separately. This adaptability represents a key breakthrough in this research, resulting in plans to continue improving the model in order to make it more accurate and reduce errors in the future.

## 3. Results and Discussion

Four dynamic system identification models were studied in the context of an automated chicken manure composting plant using forced aeration under extreme high-mountain humidity and temperature conditions. The composting process progressed satisfactorily, as shown in Figure 5. The mesophilic phase, dominated by mesophilic microorganisms breaking down soluble and degradable molecules, lasted 7 days (25–40 °C), with rapid temperature increases from spontaneous heat release. The thermophilic phase followed, where thermophilic microorganisms facilitates rapid decomposition of proteins, fats, and complex carbohydrates, lasting 5 days at approximately 45 °C. The maturation phase began on day 13, reaching 30 °C by day 16 and cooling toward ambient

temperature [1]. Physicochemical properties (e.g., C/N ratio, pH) were not measured due to experimental constraints. According to the literature, a carbon-to-nitrogen (C/N) ratio below 20 and pH range of 6.5–8.0 are indicative of mature compost [5]. The experiment was conducted in a research facility located in a high-mountain climate; this environmental condition influenced the thermophilic phase, which exhibited a moderate temperature peak below the typical range of (50–65 °C). The temperature dynamics highlight the influence of lower temperatures on heat retention. In real-world composting systems, this may require supplemental strategies such as structural insulation, longer aeration cycles, and robust heat-exchanging processes; however, as reported in composting studies, a range of (40–65 °C) is adequate for a thermophilic phase [51,52].

The dynamic responses of the BT obtained from the three modeling techniques in comparison with the experimental data are shown in Figure 5. In addition, Figure 6 hows how the temperature profile follows the expected biological behavior from the initial mesophilic stage, through the thermophilic peak, then into mesophilic cooling and maturation stage. The three models describe the basic temperature pattern of the composting process. The state-space model (mp1) demonstrated superior fitting performance (85.71%) during the mesophilic and thermophilic phases, mirroring the peaks of biological activity commonly found during proliferation of microorganisms. State-space models can represent higher-order physical systems and analyze any nonlinear systems with multiple inputs and outputs [53], making them better suited for modeling transient dynamics that occur in the early stages of composting. However, mp1 showed limitations in describing the mesophilic cooling phase, reflecting its reduced sensitivity to the decrease in metabolic activity and system stabilization. On the other hand, the process model (P2U1) showed better fitting (84.48%) during the cooling phase, although it showed lower accuracy during the biologically active phases. This suggests that this model more effectively depicts the slower dynamics associated with microbial decline and humification processes. The transfer function model (mp2) presented the lowest overall performance (55.59%), failing to model transitional behaviors or the thermophilic peak. This finding suggests that a simplified model may be insufficient to accurately represent the thermal dynamics of the composting process.

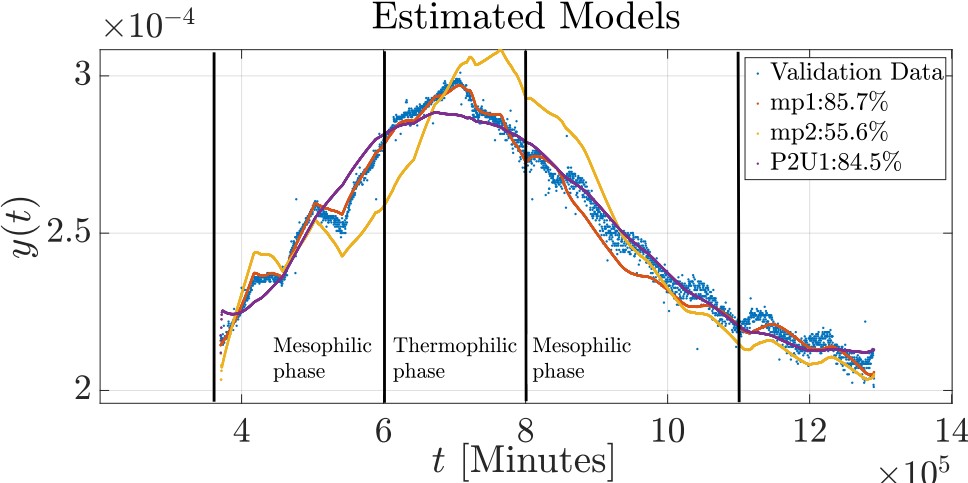

**Figure 5.** Validation of the mp1, mp2, and P2U1 models for simulating the composting process in Experiment 1.

In contrast, Figure 6 shows the BT of the biopile using the nonlinear structure, which exhibits a good model fit. However, the HW model (mhw1) exhibits oscillations throughout all phases of the composting process when compared to the validation data, and a

mismatch is observed in the initial conditions. This mismatch may be attributed to the high variability in the initial composting conditions, such as moisture content, material density, and microbial diversity. Nevertheless, the model provided a more accurate representation during the mesophilic cooling phase. Similarly, Figure 7 displays the validation results for all models using 50% of the real data that were not used in the estimation process. All models reproduce the general composting behavior to some extent.

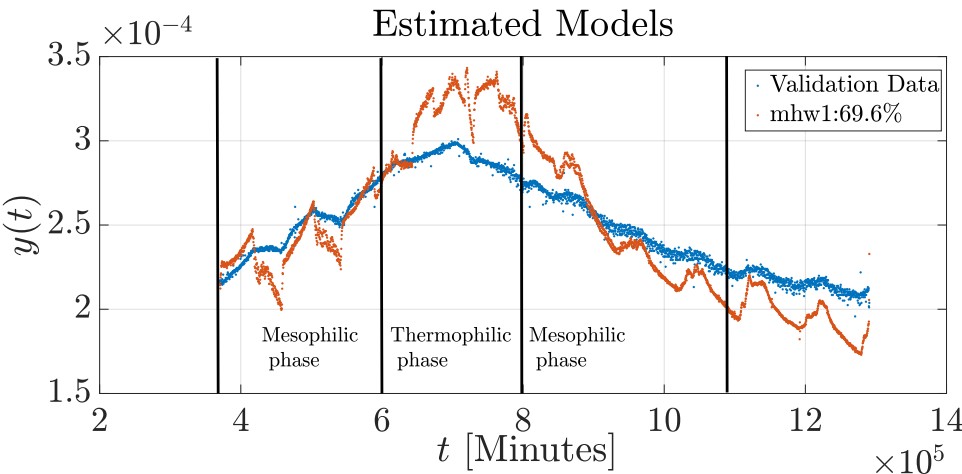

**Figure 6.** Validation of the nonlinear simulation model of the composting process in Experiment 1.

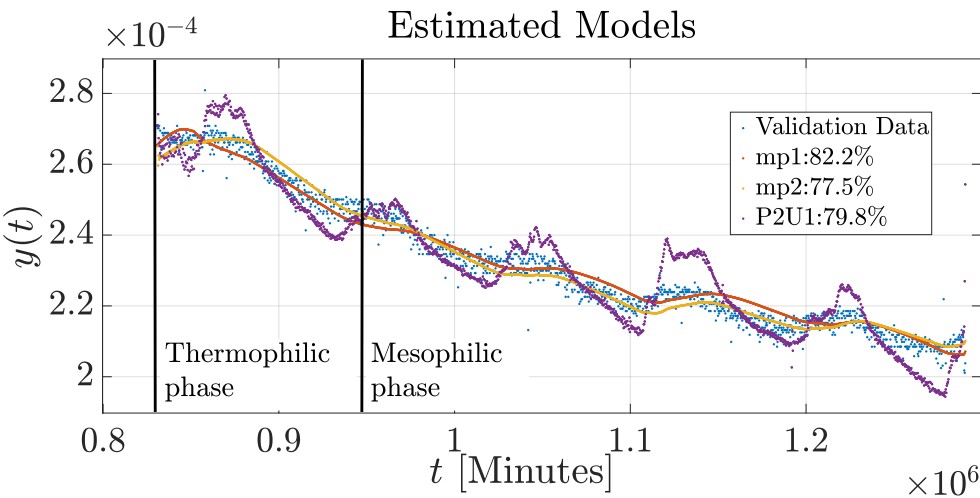

**Figure 7.** Validation of all simulation models of the composting process.

Subsequently, the results of the four model structures were compared using data from Experiment 2, which were different from those used in the identification process. Figure 8 illustrates the behavior of three models compared against the data from Experiment 2. The mp1 and P2U1 models were able to reproduce the basic dynamics of the composting process. In contrast, the mp2 model failed to reflect the fundamental temperature profile, exhibiting a significant mismatch. The mp1 model also showed deviations in the initial conditions as well as during the mesophilic and thermophilic phases. The P2U1 model demonstrated improved performance in capturing the initial conditions and the mesophilic phase compared to the mp1 model; however, it was not able to accurately reproduce the behavior during the thermophilic phase.

Figure 9 presents the results of the model with a nonlinear structure using data from Experiment 2. A noticeable discrepancy is observed in the initial conditions and

across all phases of the composting process. To determine the fitness and error of the models, we considered several well-known metrics: the normalized root mean square error (NRMSE), expressed as $NMSE = \frac{\sum_{i=1}^{n}(y_i - \hat{y}_i)^2}{\sum_{i=1}^{n}(y_i - \bar{y})^2}$; the root mean square error (RMSE), $RMSE = \sqrt{\frac{1}{n}\sum_{i=1}^{n}(y_i - \hat{y}_i)^2}$; and the mean absolute error (MAE), $MAE = \frac{1}{n}\sum_{i=1}^{n}|y_i - \hat{y}_i|$, which indicates the overall quality of the adjustment. The performance metrics computed for the different estimated models are summarized in Table 2.

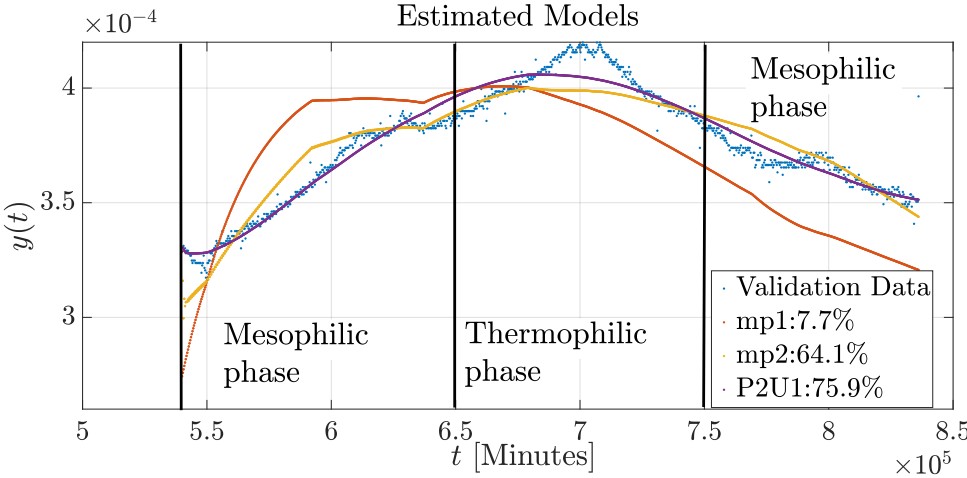

**Figure 8.** Validation of the mp1, mp2, and P2U1 simulation models of the composting process in Experiment 2.

The system identification approach effectively modeled poultry litter composting under high-mountain climate conditions, with the P2U1 process model achieving the highest fit (75.89% in Experiment 2). The inaccuracies observed during the thermophilic phase may be attributed to the biological complexity of the process, as microbial activity is highly sensitive to variations in composting parameters such as aeration and moisture content. These variations exhibit the need for adaptive modeling strategies that can effectively address environmental variability and emphasize the importance of integrating real-time environmental sensing. Overall, our study demonstrates the potential of using system identification as a data-driven tool for optimizing composting processes in Cundinamarca, Colombia without relying on complex biochemical models [7].

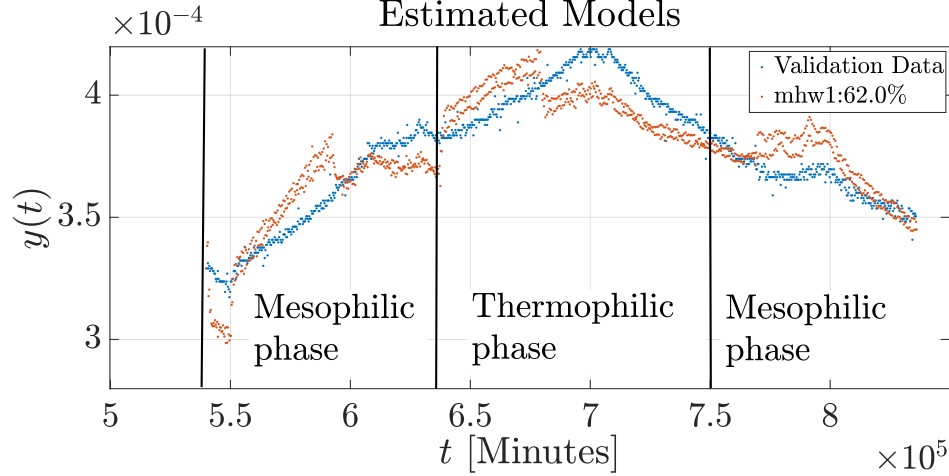

**Figure 9.** Validation of the nonlinear simulation model of the composting process in Experiment 2.

The use of system identification to effectively model poultry litter composting in the high-mountain climate of Cundinamarca, Colombia offers a data-driven alternative to complex biochemical models. Table 2 presents the comparative results of the dynamic models evaluated under three experimental conditions. In Experiment 1, using 100% of the data, the mp1 model achieved the highest fit (85.71%) and the lowest error values (RMSE: 0.581 °C, MAE: 0.432 °C), accurately capturing temperature dynamics under ideal data conditions. The P2U1 model also performed well (fit: 84.4%), with slightly higher errors but consistent accuracy. In contrast, the mp2 and mhw1 models demonstrated lower performance, with reduced fits (55.59% and 69.63%, respectively) and higher error magnitudes.

**Table 2.** Model fit and error metrics.

| Models | Experiment 1 100% Data | | | Experiment 1 50% Data | | | Experiment 2 100% Data | | |
|---|---|---|---|---|---|---|---|---|---|
| | Fit (%) | RMSE (°C) | MAE (°C) | Fit (%) | RMSE (°C) | MAE (°C) | Fit (%) | RMSE (°C) | MAE (°C) |
| mp1 | 85.71 | 0.58134 | 0.43198 | 82.20 | 0.49413 | 0.39382 | 7.70 | 3.3404 | 2.9945 |
| mp2 | 55.59 | 1.8067 | 1.5148 | 77.53 | 0.62359 | 0.49329 | 64.11 | 1.2990 | 0.9837 |
| mhw1 | 69.63 | 1.2354 | 1.0001 | 79.78 | 0.5613 | 0.41874 | 62.05 | 1.3736 | 1.1424 |
| P2U1 | 84.48 | 0.6312 | 0.52305 | 84.45 | 0.43174 | 0.33555 | 75.89 | 0.87271 | 0.64844 |

When evaluated using only 50% of the data in Experiment 1, simulating sensor failure or data scarcity, the P2U1 model outperformed all others (fit: 84.45%, RMSE: 0.432 °C, MAE: 0.336 °C), demonstrating greater robustness and adaptability. Although the mp1 and mhw1 models maintained acceptable performance, with fits of 82.20% and 79.78%, respectively, their error metrics increased slightly, indicating reduced reliability under limited data conditions. Interestingly, the mp2 model showed improved performance in this case, but still did not surpass P2U1, particularly in overall accuracy and consistency.

In Experiment 2, where process variability was significantly higher, model performance declined across the board. The mp1 model's fit dropped sharply to 7.70% (RMSE: 3.340 °C, MAE: 2.995 °C), highlighting its limited adaptability. In contrast, the P2U1 model again led the results with a strong fit of 75.89% and relatively low errors (RMSE: 0.873 °C, MAE: 0.648 °C), confirming its ability to handle dynamic and uncertain composting conditions. The mhw1 and mp2 models showed intermediate performance but did not match the robustness of P2U1. These findings suggest that simpler models such as mp1 perform well under stable and complete data conditions but struggle under variable or incomplete scenarios; conversely, P2U1 remains accurate and reliable across a range of operating conditions. Although it exhibited some limitations during the thermophilic phase, the adaptability of a model to process variability is more critical for composting systems located in high-mountain climates than achieving extreme accuracy in a specific phase. Overall, these results demonstrate the potential of P2U1 for practical composting applications, and reinforce system identification as a promising tool for optimizing processes in unpredictable environments.

Model performance was evaluated not only through fit and error metrics but also through statistical residual analysis, specifically:

- The whiteness test; a good model has its residuals within the confidence interval of the data obtained from the simulation model, with the exception of *lag* 0, which ensures that the residuals are uncorrelated.
- The test of independence; a good model has residuals that are uncorrelated with the data, i.e., the residuals should not show a systematic pattern.

Figure 10 presents the autocorrelation of the residuals and the cross-correlation between the residuals and the inputs based on the validation using 50% of the data from Experiment 1. It is evident that the autocorrelation for the mp2 (red) and mhw1 (light blue) models is the highest among all cases. A similar trend is observed in the cross-correlation results, indicating that the residuals of these two models are significantly correlated with the input data, which is an indication of poor model quality. The shaded band along the x-axis represents the 99% confidence interval; correlations falling within this region are considered statistically insignificant, provided that the lag values remain close to zero. The figure illustrates that the residuals of the mp2 and mhw1 models exceed the confidence bounds more frequently, suggesting the presence of structure in the residuals that these models fail to capture. In contrast, the state-space model (mp1) (green) and process model (P2U1) (purple) exhibit residual behaviors that fall mostly within the confidence region, further supporting the better performance and adequacy of these models seen in the previous evaluations.

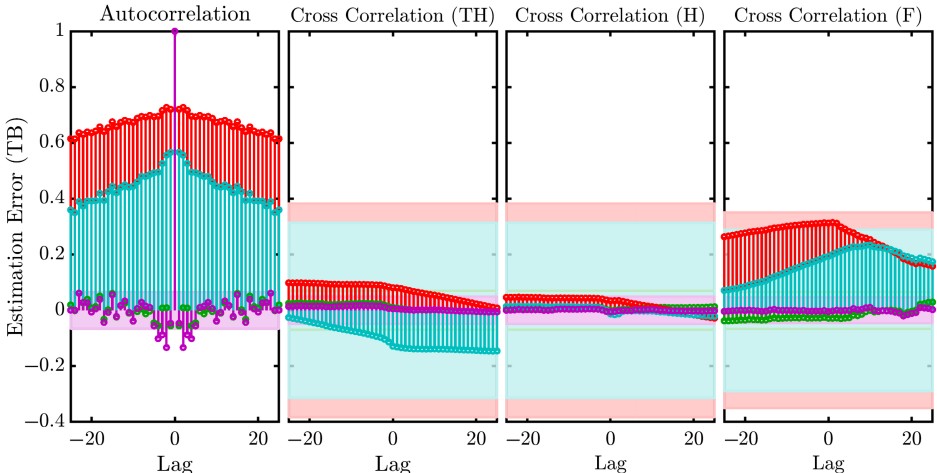

**Figure 10.** Residuals analysis, showing autocorrelation of the models with data from Experiment 1.

Finally, Figure 11 shows the autocorrelation of the residuals and cross-correlation between the residuals and inputs for model validation using 100% of the data from Experiment 2. The P2U1 process model and HW model exhibit the highest autocorrelation (red and light blue, respectively); in contrast, the Hammerstein–Wiener and state-space models show higher cross-correlation than the mp1 and P2U1 models. The P2U1 model shows the best fit to the experimental data. Composting processes involving microbial activity, heat transfer, and organic matter degradation are complex due to variable feedstock, making precise modeling challenging. Mechanistic models are often inflexible and data-intensive, making them less suitable for decision-making than the data-driven approach provided by system identification, which enhances accessibility and optimization [54]. The P2U1 model's robust fit (75.89%) supports its integration into the IoT system (Figure 2) by embedding its transfer function in a Raspberry Pi controller to adjust fan speed or heat exchanger settings for thermophilic conditions. Based on simulations, the model predicts potential aeration energy savings of 10–15%; however, these are simulation-based estimates and require real-world validation through operational testing in the pilot plant [23].

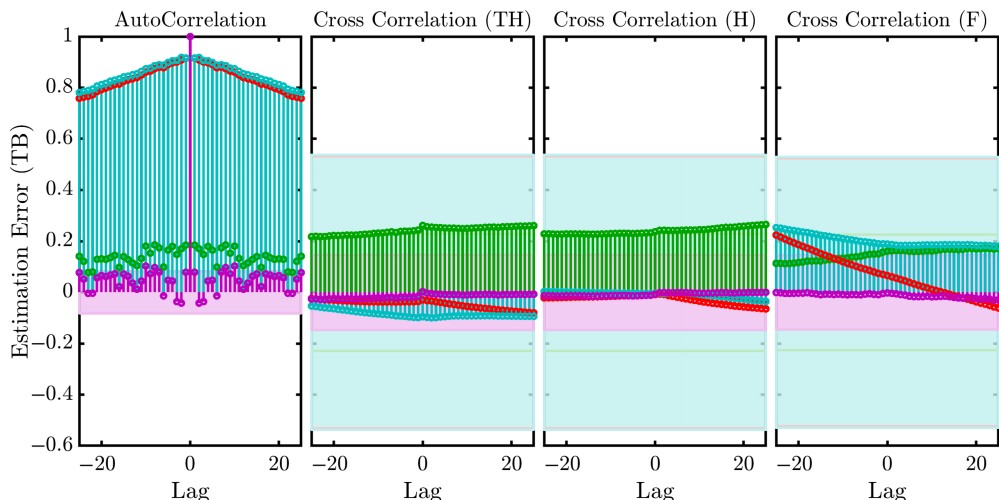

**Figure 11.** Residuals analysis, showing autocorrelation of the models with data from Experiment 2.

## 4. Conclusions

This study employed system identification to model the composting process in forced-aeration piles under high mountain climates utilizing four techniques: three linear models (state-space, process, and transfer function) and one nonlinear model (Hammerstein–Wiener). The process model best captured the mesophilic phases, while the state space model excelled in describing both mesophilic and thermophilic stages. Despite capturing mesophilic behavior, the HW model showed limitations due to oscillations, while the transfer function model underperformed overall. Validation with independent datasets revealed the process model's superior predictive accuracy, although none of the models adequately described the thermophilic phase, likely due to environmental variability. This study's key contribution lies in demonstrating system identification as a robust data-driven alternative to traditional energy and mass balance approaches, enabling accurate modeling of composting dynamics without extensive biochemical characterization. However, limitations include the tested models' focus on temperature, excluding moisture and ventilation, and the use of identical substrates, which restricts generalizability. Future work should integrate these factors and explore adaptive or hybrid models in order to enhance robustness and scalability.

This study provides practical ideas for sustainably handling waste in poultry farming. Using the tested IoT setup, farmers can monitor key factors such as airflow and temperature in real time, which helps to save energy and produce better compost. What stands out is how flexible this method is; for instance, it can be adjusted to work with various materials and climates, meaning that it is not limited to high-mountain areas. Finally, these models can be easily integrated into existing composting processes, taking into account automated tools that fit into modern agricultural systems. This approach enables informed decisions, can predict when compost is ready with greater accuracy, and considers environmental regulations, helping to make farming more sustainable.

**Author Contributions:** Conceptualization, A.A.P.-F. and H.R.-M.; methodology, F.S.-C. and and F.F.F.; software, A.A.P.-F. and F.S.-C.; validation, R.R. and G.S.-R.; formal analysis, F.S.-C. and H.R.-M.; investigation, R.R. and F.F.F.; data curation, H.R.-M. and R.R.; writing—original draft preparation, F.S.-C. and H.R.-M.; writing—review and editing, R.R. and G.S.-R.; funding acquisition, A.A.P.-F. All authors have read and agreed to the published version of the manuscript.

**Funding:** This research was funded by Vicerrectoría de Investigación y Transferencia (VRIT) at Universidad de La Salle, Bogotá, Colombia under institutional code IAU202-146. The APC was funded by Universidad de La Salle, Bogotá, Colombia.

**Institutional Review Board Statement:** Not applicable.

**Informed Consent Statement:** Not applicable.

**Data Availability Statement:** Datasets related to this article can be found at https://data.mendeley.com/datasets/dgxxj2pk8s/2 (accessed on 25 June 2025), hosted at Mendeley [39].

**Conflicts of Interest:** The authors declare no conflicts of interest.

## Abbreviations

The following abbreviations are used in this manuscript:

| | |
|---|---|
| AI | Artificial Intelligence |
| ANN | Artificial Neural Networks |
| ARMA | Autoregresive Moving Average |
| BT | Biopile Temperature |
| CatBoost | Categorical Boosting |
| C/N | Carbon-to-Nitrogen ratio |
| ET | Extra Trees |
| GI | Germination Index |
| GPRS | General Packet Radio Service |
| GSM | Global System for Mobile Communications |
| H | Humidity |
| HI | Humification Index |
| HT | Heater Temperature |
| HW | Hammerstein–Wiener |
| IoT | Internet of Things |
| LSI | Latent Semantic Indexing |
| MAE | Mean Absolute Error |
| mhw1 | Hammerstein–Wiener Model |
| MISO | Multiple-Input–Single-Output |
| ML | Machine Learning |
| mp1 | State-Space Model |
| mp2 | Transfer Function Model |
| NRMSE | Normalized Root Mean Square Error |
| OE | Output-Error Polynomial Algorithm |
| P | Process |
| P2U1 | Process Model |
| RF | Random Forest |
| RMSE | Root Mean Square Error |
| RSM | Response Surface Methodology |
| SS | State Space |
| TF | Transfer Function |
| XGBoost | eXtreme Gradient Boosting |

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
