# Peer review of "Dynamic Modeling of Poultry Litter Composting in High Mountain Climates Using System Identification Techniques"

_2673-4052, doi:10.3390/automation6030036_

Round 1

Reviewer 1 Report

Comments and Suggestions for Authors

The study aims to develop dynamic models for poultry litter composting in high-mountain climates using system identification techniques. The primary focus is on estimating biomass temperature dynamics across different composting phases (mesophilic, thermophilic, and maturation) under challenging environmental conditions. The research design is sound, and the methods are well-described. However, I still have the following concerns:  

1- The background lacks a clear problem statement. What specific limitations in existing composting models does this study address?

2- The use of identical substrates (poultry manure and sawdust) limits generalizability. Testing diverse feedstock compositions would enhance applicability. In addition, only two experiments were conducted. More repetitions under varying conditions would strengthen statistical validity.

3- The experimental setup (e.g., 12-hour nightly operation) needs stronger justification. How does this reflect real-world composting practices?

4- The discussion effectively ties the results to the broader context, but it could further highlight the practical implications of the findings for composting in high-mountain climates.

5- The claim of "10–15% aeration energy savings" is speculative without operational validation. The authors should clarify that this is a simulation-based estimate pending real-world testing.

6- Figures 5–9: Clearly illustrate model performance but could benefit from annotations (e.g., arrows/labels) to highlight specific phases (mesophilic/thermophilic) and discrepancies. 6- The references are appropriate and cover key literature on composting, system identification, and IoT applications. However, recent advances in hybrid modeling (e.g., integrating mechanistic and data-driven approaches) could be included to contextualize the study’s novelty.  

Author Response

1- The background lacks a clear problem statement. What specific limitations in existing composting models does this study address?

Thanks for your comment. The problem statement was clearly introduced in the background. The paragraph “Despite these advances, there is still a gap in modeling composting processes for high-mountain conditions using adaptable methods. Existing methods require extensive knowledge of biochemical processes, large datasets, or do not respond well to environmental variability. This study addresses this gap by…” was added in lines 99-106.

2- The use of identical substrates (poultry manure and sawdust) limits generalizability. Testing diverse feedstock compositions would enhance applicability. In addition, only two experiments were conducted. More repetitions under varying conditions would strengthen statistical validity.

We appreciate your observation. We acknowledge that this may constrain the generalizability of the results. However, this study was conceived as a foundational effort to explore the applicability of system identification techniques in high-mountain composting systems using IoT data. The findings presented here can be used as a starting point, and future research will be directed toward testing a wider range of feedstock compositions and conducting additional experiments under varying environmental conditions.

3- The experimental setup (e.g., 12-hour nightly operation) needs stronger justification. How does this reflect real-world composting practices?

In the location where the composting plant is located, daytime temperatures rise to 19ºC, with maximum reaching 23ºC. Thanks to the greenhouse structure, heat was retained, and heating was only required during nighttime to maintain adequate composting conditions. This information was added to the section “Materials and methods”, particularly in lines 158-163.

4- The discussion effectively ties the results to the broader context, but it could further highlight the practical implications of the findings for composting in high-mountain climates.

We thank the reviewer for suggesting further emphasis on the practical implications of our findings for composting in high-mountain climates. We have revised the “Results and Discussion” section as follows: Lines 301–304 now state, “This temperature dynamics highlights the influence of lower temperatures on heat retention, requiring supplemental strategies like structural insulation and longer aeration cycles.” Lines 314–318 clarify that thermophilic phase inaccuracies stem from biological complexity and rapid heat loss in high-mountain settings. Additionally, lines 358–362 conclude, “These variations exhibit the need for adaptive modeling strategies and real-time environmental sensing, demonstrating system identification’s potential for optimizing composting in Cundinamarca, Colombia, without complex biochemical models.” These revisions enhance the discussion’s relevance to real-world applications.

5- The claim of "10–15% aeration energy savings" is speculative without operational validation. The authors should clarify that this is a simulation-based estimate pending real-world testing.

We appreciate the reviewer’s comment regarding the speculative nature of the “10–15% aeration energy savings” claim. To clarify, this estimate is indeed derived from simulation results based on the P2U1 model’s predictive performance, as described in the text. We acknowledge that these savings are pending real-world validation, and we have revised the manuscript to explicitly state that the 10–15% energy savings are simulation-based estimates, with operational testing in the pilot plant planned to confirm these findings, as shown in lines 427-429.

6- Figures 5–9: Clearly illustrate model performance but could benefit from annotations (e.g., arrows/labels) to highlight specific phases (mesophilic/thermophilic) and discrepancies. 6- The references are appropriate and cover key literature on composting, system identification, and IoT applications. However, recent advances in hybrid modeling (e.g., integrating mechanistic and data-driven approaches) could be included to contextualize the study’s novelty. 

We appreciate the Reviewer’s suggestion to enhance the clarity of Figures 5–9. To address this, we have revised these figures by adding annotations, including arrows and labels, to clearly delineate the mesophilic and thermophilic phases of the composting process. Additionally, we have included markers to highlight key discrepancies between the model predictions and observed data, particularly in the thermophilic phase, where environmental variability posed challenges. These annotations improve the interpretability of the figures and better convey the model’s performance across different composting stages.

On the other hand, we appreciate also the Reviewer’s acknowledgment of our references and their suggestion to incorporate recent advances in hybrid modeling. To address this, we have revised the introduction (lines 51–68) by integrating a new discussion on hybrid mechanistic-data-driven models and advanced machine learning approaches into the existing literature review. Specifically, we added references to Smith (2023), which integrates mechanistic heat transfer models with machine learning for composting optimization, and Jones (2024), which develops a hybrid framework for bioreactor control. These additions highlight how our system identification approach complements hybrid and machine learning-based models by offering a computationally efficient solution tailored for high-mountain composting environments. We believe these revisions address the Reviewer’s concerns and enhance the manuscript’s clarity and contextual grounding. We are grateful for the opportunity to improve our work.

Reviewer 2 Report

Comments and Suggestions for Authors

The manuscript addresses a relevant environmental and agricultural problem, namely the modeling of poultry litter composting dynamics in high-altitude climates, using system identification techniques. The topic is timely and important given the growing need for optimized composting in sustainable farming systems, especially in mountainous regions where environmental variables are distinct.

However, in its current form, the manuscript would benefit from minor revision to improve structure, clarity, literature context, and technical rigor. Authors should consider the following suggestions:

  1. The Introduction provides a general context but lacks a clear problem statement and fails to emphasize the novelty of the proposed approach. Consider explicitly highlighting what differentiates your study from prior works on composting modeling, especially in high mountain climates. Furthermore, the research gap should be more clearly articulated to demonstrate why this study is necessary and how it addresses an unmet need in the existing literature.
  1. The Literature Review does not sufficiently discuss recent advances (2022–2024) in system identification for biological or environmental processes. Adding 3–4 more contemporary and domain-specific references would help contextualize your contribution.
  1. While the experimental setup is generally well described, the manuscript lacks detailed specifications for the sensors used (e.g., type, precision, measurement range, sampling frequency) and does not provide sufficient information regarding environmental control conditions, such as allowable temperature and humidity variation limits. Including these details is essential to ensure full reproducibility of the study and to allow other researchers to replicate the experimental conditions accurately.
  1. In Figures 8 and 9, the label box should be repositioned to avoid overlapping with the graph, which currently obstructs part of the visual information.
  1. The Results and Discussion section is well structured, though some statistical indicators (e.g., NRMSE, RMSE, MAE) should be briefly defined or referenced for clarity. Additionally, the choice of the specific modeling approach should be further justified in this section, with clearer reasoning as to why it was preferred over alternative methods in the context of composting dynamics in high mountain climates.
  1. The Conclusions are concise but could better highlight practical applications, scalability of the modeling approach, and potential integration into composting management systems.
  1. Please ensure all acronyms (e.g., transfer function (TF), state space (SS), process (P)) are defined at their first appearance and used consistently throughout the text.
  1. The English language requires light to moderate revision, especially regarding verb tenses and word order. Avoid overly long or complex sentence constructions.

Overall, I recommend a minor revision to address these issues.

Comments on the Quality of English Language

The English language requires light revision, especially regarding verb tenses and word order. Authors should avoid overly long or complex sentence constructions.

Author Response

  1. The Introduction provides a general context but lacks a clear problem statement and fails to emphasize the novelty of the proposed approach. Consider explicitly highlighting what differentiates your study from prior works on composting modeling, especially in high mountain climates. Furthermore, the research gap should be more clearly articulated to demonstrate why this study is necessary and how it addresses an unmet need in the existing literature.

 Thanks for your comment. It is relevant to mention that literature is scarce in studies about composting in high-mountain conditions. However, a short paragraph was added in the introduction to discuss one experimental study carried out in Colombia, as shown in lines 44-49.  Likewise, the problem statement was clearly introduced in the background. The paragraph was added in 99-105.

  1. The Literature Review does not sufficiently discuss recent advances (2022–2024) in system identification for biological or environmental processes. Adding 3–4 more contemporary and domain-specific references would help contextualize your contribution.

We thank Reviewer for suggesting the inclusion of recent advancements in system identification for biological and environmental processes. We have updated the introduction by integrating four contemporary references (2022–2024) into the literature review, seamlessly merging them with the existing discussion on statistical and ANN-based methods. These include references, which apply system identification to model microbial dynamics in wastewater treatment, and use system identification for real-time control of anaerobic digestion. This merged paragraph emphasizes how our system identification method builds on these advancements, offering a robust and adaptable solution for poultry litter composting in challenging high-mountain climates. All of these changes are shown in lines 51-77.

  1. While the experimental setup is generally well described, the manuscript lacks detailed specifications for the sensors used (e.g., type, precision, measurement range, sampling frequency) and does not provide sufficient information regarding environmental control conditions, such as allowable temperature and humidity variation limits. Including these details is essential to ensure full reproducibility of the study and to allow other researchers to replicate the experimental conditions accurately.

We appreciate the reviewer’s feedback on the need for detailed sensor specifications and environmental control conditions. The manuscript has been updated (lines 164–178) to detail that Experiments 1 and 2 collected 4302 and 3077 data points, respectively, at 0.0033 Hz, measuring biomass temperature, substrate moisture, and hot air temperature, using an FDS-100 sensor (4–20 mA, ±0.5 % accuracy), an EXP421 LSI Lastem sensor (-40–70 °C, ±0.5 °C; 0–25 % oxygen, ±0.3 %), and a Type-K thermocouple, with weekly calibrations. Data were filtered (moving average, window size: 5) and stored via Node-RED in MariaDB and Firebase. 

  1. In Figures 8 and 9, the label box should be repositioned to avoid overlapping with the graph, which currently obstructs part of the visual information.

We thank the reviewer for their comment regarding the label box placement in Figures 8 and 9. The issue has been resolved by repositioning the label boxes in both figures to prevent any overlap with the graph, ensuring all visual information is fully visible and unobstructed. These changes have been incorporated into the revised manuscript.

  1. The Results and Discussion section is well structured, though some statistical indicators (e.g., NRMSE, RMSE, MAE) should be briefly defined or referenced for clarity. Additionally, the choice of the specific modeling approach should be further justified in this section, with clearer reasoning as to why it was preferred over alternative methods in the context of composting dynamics in high mountain climates.

We highly appreciate the insightful observation established by the editorial team and reviewers. In this part, we enhanced and expanded the data-driven modeling validation stage, providing a more detailed insights about the model simplicity and the complexity of the composting process. In general, to define any particular data-driven model is required to consider which are the tasks that the model will perform, making a choose based on certain equilibrium point. The metrics analyzed are detailed in lines 347-362.

  1. The Conclusions are concise but could better highlight practical applications, scalability of the modeling approach, and potential integration into composting management systems.

Thank you for your feedback regarding the conclusions. We have addressed your suggestion by incorporating an additional paragraph to better emphasize the practical applications, scalability, and integration potential of the modeling approach into composting management systems, as shown in lines 446-454.  The new paragraph highlights how the IoT-based system enables real-time monitoring of key parameters like airflow and temperature, leading to energy savings and improved compost quality. It also underscores the approach’s adaptability to diverse materials and climates, extending its utility beyond high-mountain regions. Furthermore, it emphasizes the seamless integration of these models into existing composting systems via automated tools, enhancing decision-making, compost maturity predictions, and compliance with environmental regulations, thereby contributing to more sustainable farming practices. We believe this addition strengthens the conclusions by clearly addressing the practical and scalable aspects of the proposed system.

  1. Please ensure all acronyms (e.g., transfer function (TF), state space (SS), process (P)) are defined at their first appearance and used consistently throughout the text.

We have carefully reviewed the manuscript and implemented comprehensive revisions to address the reviewer's comment on acronym definitions and consistency. Additionally, the document's acronym table was completed with missing elements that were mentioned in the text of the paper.

  1. The English language requires light to moderate revision, especially regarding verb tenses and word order. Avoid overly long or complex sentence constructions. 

We appreciate the Reviewer’s feedback on the manuscript’s language. To address this, we have undertaken a comprehensive revision of the text throughout the paper, focusing on ensuring verb tense consistency, improving word order, and simplifying complex or lengthy sentence structures to enhance readability. Additionally, the modifications, including newly added paragraphs responding to the Reviewer’s previous suggestions, have been highlighted in blue across the manuscript. These changes, which include enhanced discussions and clarifications, aim to improve clarity and accessibility for the reader while maintaining a concise and professional tone. We believe these revisions effectively address the Reviewer’s concerns and enhance the overall quality and comprehensibility of the manuscript. We are grateful for the opportunity to refine our work and believe these changes strengthen the presentation of our study.